# Determinants and predictors of mental health during and after COVID-19 lockdown among university students in Malaysia

**Imtiyaz Ali Mir**[1][☯]*, **Shang Kuan Ng**[1][☯], **Muhammad Noh Zulfikri Mohd Jamali**[1], **Mohammed AbdulRazzaq Jabbar**[2], **Syeda Humayra**[3]

1 Department of Physiotherapy, M Kandiah Faculty of Medicine and Health Sciences, Universiti Tunku Abdul Rahman, Jalan Sungai Long, Selangor, Malaysia, 2 Department of Population Medicine, M Kandiah Faculty of Medicine and Health Sciences, Universiti Tunku Abdul Rahman, Jalan Sungai Long, Selangor, Malaysia, 3 Department of Public Health, Faculty of Allied Health Sciences, Daffodil International University, Dhaka, Bangladesh

☯ These authors contributed equally to this work.
* imtiyaz2204@gmail.com

**Data Availability Statement:** All relevant data are within the paper and its Supporting information files.

## Abstract

### Background

Young adults, particularly university students might be at greater risk of developing psychological distress, and exhibiting symptoms of anxiety and depression during the COVID-19 pandemic. The primary objective of this study was to explore and compare the determinants and predictors of mental health (anxiety and depression) during and after the COVID-19 lockdown among university students.

### Methods

This was an observational, cross-sectional study with a sample size of 417 students. An online survey utilizing International Physical Activity Questionnaire–Short Form (IPAQ-SF), General Anxiety Disorder–7 (GAD-7) and Patient Health Questionnaire-9 (PHQ-9) was distributed to Universiti Tunku Abdul Rahman students via Google forms.

### Results

During lockdown, family income [$\chi^2$ (1, n = 124) = 5.155, p = 0.023], and physical activity (PA) [$\chi^2$ (1, n = 134) = 6.366, p = 0.012] were associated with anxiety, while depression was associated with gender [$\chi^2$ (1, n = 75) = 4.655, p = 0.031]. After lockdown, family income was associated with both anxiety [$\chi2$ (1, n = 111) = 8.089, p = 0.004], and depression [$\chi2$ (1, n = 115) = 9.305, p = 0.002]. During lockdown, family income (OR = 1.60, p = 0.018), and PA (OR = 0.59, p = 0.011) were predictors for anxiety, while gender (OR = 0.65, p = 0.046) was a predictor for depression. After lockdown, family income was a predictor for both anxiety (OR = 1.67, p = 0.011), and depression (OR = 1.70, p = 0.009).

**Funding:** The funders had no role in study design, data collection and analysis, decision to publish, or preparation of the manuscript.

**Competing interests:** The authors have declared that no competing interests exist.

## Conclusion

Significant negative effects attributed to the COVID-19 lockdown, and certain factors predisposed to the worsening of mental health status in university students. Low family income, PA, and female gender were the major determinants and predictors linked to anxiety and depression.

## Introduction

The COVID-19 pandemic has affected the entire globe at a high magnitude, causing unprecedented consequences, and possibly altering the way we exist in nature. This catastrophe has solidified itself as one of the worst crises that have ever existed in human history, one that not only affects an individual's health but also disrupts the social and economic harmony of society. COVID-19, in essence, is a respiratory disease, with common infection symptoms such as fever, chills, cough, shortness of breath, fatigue, etc., which is airborne and highly contagious [1]. To curb the infection, various public health measures were implemented, including the Movement Control Order (MCO), where social and religious gatherings, outdoor recreational activities, operation of learning institutions of all levels were prohibited, and only essential stores and health care facility centres were allowed to operate [2, 3]. Previous studies have shown that quarantine brought adverse mental health effects to people even after the lockdown period. The aftereffects of quarantine during the SARS outbreak suggested that worrying negative consequences may continue for prolonged periods [4, 5].

Young adults, particularly university students, are at greater risk of psychological distress in case of a health emergency, as supported by findings where approximately 36% and 73% of the sample exhibited symptoms of anxiety and depression during the COVID-19 pandemic [6]. This seems consistent with the findings from French, British, and American studies conducted among university students during the COVID-19 lockdown [7–9]. A recent study in Northern Italy pointed out the detrimental effects of social isolation and confinement in relation to university students' lifestyle habits, behavioural risk factors, and highlighted its impact on the psychosomatic system. The length of isolation caused a higher incidence of headaches, gastro-intestinal issues, sadness, depression, and COVID-19 fear. The risk of health outcomes was higher in females and linked to sleep quality, memory difficulties, performance decline, and medium-intensity phone use [10]. Another study that recruited participants from 20 countries found that post-traumatic stress disorder (PTSD) caused by COVID-19 was significantly associated with depression and anxiety; 69% of participants experienced depression, while 57% had anxiety symptoms [11]. During a mere 2 weeks of COVID-19 quarantine, 45% of 530 students in a survey strongly agreed that they felt emotionally detached from their closed ones, and a quarter of them felt depressed [12]. Along with pre-existing stressors such as increased academic load, peer pressure, parental pressure, and future uncertainty, the students have been coping with social distancing, personal protection from the pandemic, overburdened responsibilities, and inadequate resources [13, 14]. In addition, they have also been experiencing disengagement in meaningful activities, which may offer positive reinforcement in their academic journey [13]. Thus, university students are possibly one of the most vulnerable population-, as they abruptly switched to online learning while waiting for the reopening of institutions since they have to apply the skills they learned at university to kick-start their career after graduation [15]. This sudden transition from traditional classroom learning to e-learning is not a pleasant experience and is significantly challenging for many students. Hence it is

paramount to investigate the factors that could influence the students' satisfaction with e-learning during an emergency crisis like COVID-19 to alleviate their psychological distress [16, 17].

Adolescence is frequently considered the first stage of development for mental health issues like mood disorders, social anxiety, and substance abuse. Adolescent-onset mental health problems orchestrate lifelong mental health and well-being by comprehending the need to assess at-risk or vulnerable people since individual variability in brain development, its causes and consequences, and the link with mental health demand urgent consideration [18]. Being in a phase where students are on the verge of transitioning into adulthood, they need to have the optimum mental health, which allows them to be effective contributors to society in the future. It has been well-documented that during this transition, students are more prone to develop mental health problems, and those with pre-existing mental health issues are at greater risk of exaggerated symptoms [13]. People diagnosed with major depressive disorders may develop bipolar disorder and present with poor performance, substance abuse, suicidal tendencies, increased hospitalization, and involvement in legal issues [19, 20]. It could be related to the alterations in the brain, particularly in the white matter and neural circuitry, since both these structures are critically involved in sensory processing and emotion regulation [20]. In addition, the anterior cingulate cortex, insula, and dorsal striatum may differ significantly among depressed people compared to healthy controls [21]. However, it remains unclear whether these changes are the risk factor for depression or occur due to the disorder itself [22]. Previous literature has also highlighted the structural brain changes concerning the association between neurodegenerative diseases and COVID-19, in which the severity of infection, the extent of neurologic symptoms, and its consequential impact acted as the key determinants [23].

Over the past decade, there has been a sharp increase in mental illnesses in Malaysia. Due to a significant change in the educational system, particularly in the way that instruction is delivered, university students in Malaysia have been experiencing high levels of depression, anxiety, and stress amidst the COVID-19 pandemic [24]. Hence it is essential to assess the determinants of mental health of this particular population so that the appropriate counter measures can be implemented for those vulnerable in sub-optimal levels. In the Asian culture especially, showing emotions and revealing vulnerabilities are rarely practiced, which diverts students to opt for maladaptive coping strategies when they face stress and anxiety, which does not improve the underlying problem and would only worsen the situation with time [25]. Therefore, the objective of this study was to explore and compare the determinants and predictors of mental health (anxiety and depression) during and after COVID-19 lockdown among the university students in Malaysia, with the following research hypotheses: (1) COVID-19 lockdown has a significant effect on the mental health of university students; (2) there is a significant difference in the symptoms of anxiety and depression during and after the COVID-19 lockdown; (3) there is a significant difference in the determinants and predictors of mental health during and after the COVID-19 lockdown.

## Materials and methods

### 2.1 Study design and participants

A cross-sectional study design was implemented that adhered to the STrengthening the Reporting of OBservational studies in Epidemiology (STROBE) guidelines. The participants of focus were the Universiti Tunku Abdul Rahman (UTAR) students. The sampling method employed in this research was non-probability convenience sampling. Study participants were invited through university portal and Microsoft Teams subscribed under institutional emails.

An online survey was prepared utilizing the Google Forms as face-to-face interaction with participants was not possible during the pandemic period. All the important information like objective of the study, inclusion/exclusion criteria, and informed consent was provided in the cover letter of this survey. The explanation of procedure and the researcher's contact information was also placed in the first section of the survey, in case the participants required any clarification regarding the survey. The inclusion criteria were that the students must be active UTAR students, aged 18–26 years old, and of both genders, while the exclusion criteria were participants who has had any prior history of anxiety and depression before the COVID-19 pandemic or has not opted for voluntary participation. The sample size was calculated based on the 20158 number of UTAR students quoted by the Division of Admissions and Credit Evaluation of UTAR, and using the Krejcie and Morgan table [26], a sample size of 379 with an additional 38 students to account for the 10% attrition rate, which resulted in the final sample size of 417. The following sample size is considered large enough to reduce the data variability and minimise the bias that could arise from a smaller sample of the study population. To minimize the selection bias, we ensured that representative samples of the population were only active UTAR students, and they were required to enter the university's unique identity card number in the first section of the survey before they could proceed further.

## 2.2 Ethics and consent statement

This study was registered prospectively in clinicaltrials.org with the ID NTC05031988 and data was collected from 18 October 2021 to 16 December 2021. All the respondents were required to provide their written informed consent in the form of a digital signature before answering the questionnaire through Google forms. The study was conducted in accordance with the declaration of Helsinki and was approved by the Scientific and Ethical Review Committees (SERC) of UTAR under the ethical approval no. U/SERC/127/2021.

## 2.3 Variables

The target variables were age, gender, race, living area and condition during and after COVID-19 lockdown, restriction level before and after the COVID-19 lockdown, family income level, and pre-lockdown habits. Living area focused on whether the students were living in urban or rural regions, and living condition was asking if the students were living with their family members or living with others (friends or alone). Restriction level was complete restriction (practicing complete social distancing measure and not going out for any activity), or partial, or no restriction (practicing social distancing measure but only going out for essential activities or not practicing any social distancing measures). Family income level was based on the B40, M40 and T20 categories with B40 group pertaining to those reporting below RM4850 family income per month; M40 between RM4851 and RM10971 family income per month; T20 more than RM10971 family income per month [27]. The students were categorized into B40 group (low-income) or M40/T20 group (high-income). Pre-lockdown habits were a dichotomous 'yes' or 'no' response type of questions, asking the students whether they have been engaged in an active lifestyle prior to the lockdown. Presence of chronic disease was an open-ended question regarding any health conditions that has persisted for more than a month to be filled if relevant.

## 2.4 Outcome measures

Outcome measures including physical activity level (PA) and mental health (anxiety and depression) of the participants were assessed during and after the COVID-19 lockdown. PA was assessed through the short form version of International Physical Activity Questionnaire

(IPAQ-SF) featuring 7 questions on frequency and duration spend in various forms of PA over a period of one week, scores of which were categorized into (i) minimally active characterized by (a) 3 or more days of vigorous activity of at least 20 minutes per day OR (b) 5 or more days of moderate-intensity activity or walking of at least 30 minutes per day OR (c) 5 or more days of any combination of walking, moderate-intensity or vigorous intensity activities achieving a minimum of at least 600 MET-min/week; (ii) Health-enhancing physical activity (HEPA) characterized by a) vigorous-intensity activity on at least 3 days achieving a minimum of at least 1500 MET-minutes/week OR b) 7 or more days of any combination of walking, moderate-intensity or vigorous intensity activities achieving a minimum of at least 3000 MET-minutes/week, or (iii) inactive characterized by those that doesn't fulfil any of the above-mentioned criteria. A spreadsheet [28] was used to automatically process and group the PA data based on the criteria for each group. While for mental health, anxiety was assessed using the General Anxiety Disorder-7 (GAD-7) tool, and depression was measured using the Patient Health Questionaire-9 (PHQ-9). GAD-7 and PHQ-9 features 7 and 9 questions respectively, with scoring based on the 4-point Likert scale, and both questionnaires are reported to have a good internal consistency (GAD-7 = 0.92, PHQ-9 = 0.89) [29, 30]. Participants were required to score each statement on a scale of 0–3, and the total score was calculated and we categorized the students into none (0–4), mild (5–9), moderate (10–14), moderately severe (15–19) or severe ($\geq$ 20) depression; and minimal (0–4), mild (5–9), moderate (10–14), or severe (15–21) anxiety. Students with a PHQ-9 and GAD-7 score of 4 or less than 4 were categorized as not having any depression or anxiety respectively.

## 2.5 Statistical reporting

Data analysis was carried out with the IBM SPSS software statistics version 24. Descriptive analysis was performed on the depression and anxiety scores of both timelines. McNemar's test was carried out to assess the difference in scores of anxiety and depression during and after the lockdown. The assumptions were met which included i) one dichotomous dependent variable and one dichotomous independent variable, ii) mutually exclusive dependent variable, iii) sample is random. Chi-square test was run to assess the association between each demographic characteristic on the levels of anxiety and depression among the students. Logistic regression analysis was executed to analyse the predictors of anxiety and depression. Significance level for all the tests were set at $\leq$ 0.05.

A total of 442 responses were collected. However, upon further screening, 8 duplicate responses were identified, 1 participant did not provide consent, and 1 participants' data did not tally with the requirement of the questions. So, these 10 data sets were removed and statistical analysis was performed only on 432 data sets.

## Results

### 3.1 Demographics

Demographic characteristics of the participants are depicted in Table 1, except the gender based depression and anxiety level, which is shown in Table 3. In this study, all the participants were Malaysians. Majority of the students were aged between 18–21 years old (79.6%) and 20.4% were aged between 22–26 years. Notably, female participants showed a higher response rate (69%) compared to the male gender (31%). In terms of ethnicity, predominated response (97%) was received from Chinese students, whereas 3% of respondents were from Malay and Indian races.

During COVID-19 lockdown, majority of the participants were living with their families (92.1%), but after the COVID-19 lockdown was lifted, only 10.9% of respondents were living

**Table 1. Participant demographics.**

| Demographic data | Frequency (%) |
|---|---|
| **Age** | |
| 18–21 (0) | 344 (79.6) |
| 22–26 (1) | 88 (20.4) |
| **Gender** | |
| Male (0) | 134 (31.0) |
| Female (1) | 298 (69.0) |
| **Race/Ethnicity** | |
| Chinese (0) | 419 (97.0) |
| Non-Chinese (1) | 13 (3.0) |
| **Living condition during COVID-19 Lockdown** | |
| Living with friends/alone (0) | 34 (7.9) |
| Living with family members (1) | 398 (92.1) |
| Living condition after COVID-19 Lockdown | |
| Living with family members (0) | 47 (10.9) |
| Living with friends/alone (1) | 385 (89.1) |
| **Living area during Covid-19 Lockdown** | |
| Rural area (0) | 91 (21.1) |
| City area (1) | 341 (78.9) |
| **Living area after COVID-19 Lockdown** | |
| Rural area (0) | 82 (19.0) |
| City area (1) | 350 (81.0) |
| **Restriction level during COVID -19 lockdown** | |
| Complete restriction (0) | 232 (53.7) |
| Partial/No restriction (1) | 200 (46.3) |
| **Restriction level after COVID-19 lockdown** | |
| Complete restriction (0) | 48 (11.1) |
| Partial/No restriction (1) | 384 (88.9) |
| **Family income level** | |
| B40 (0) | 209 (48.4) |
| M40 (1) | 193 (44.7) |
| T20 (2) | 30 (6.9) |
| **Pre-lockdown habits** | |
| No (0) | 174 (40.3) |
| Yes (1) | 258 (59.7) |
| **Presence of chronic disease** | |
| None | 417 (96.9) |
| Asthma | 3 (0.7) |
| Ankylosing spondylitis | 1 (0.2) |
| Diabetes | 1 (0.2) |
| Headache | 1 (0.2) |
| Tetralogy of Fallout | 1 (0.2) |
| Hyperthyroidism | 2 (0.5) |
| Irritable bowel syndrome | 1 (0.2) |
| Scoliosis | 1 (0.2) |
| Sinusitis | 2 (0.5) |
| Thalassemia | 1 (0.2) |

with families. Most of the students (78.9%) were living in the urban areas during the lockdown, and it increased to 81% after the lockdown. More than half of the students (53.7%) were living in a complete restriction during the pandemic lockdown, and it drastically decreased to 11.1% after the restriction was eased. Nearly half of the research participants (48.4%) belonged to the B40 income groups, whereas 44.7% were from M40 and 6.9% from the T20 income groups. During the pre-lockdown period, 59.7% of students were living an active lifestyle, while 40.3% were physically inactive. Around 97% of participants were not having any co-morbidities, while 3 students (0.7%) had asthma, 2 students (0.5%) were diagnosed with hyperthyroidism and 2 (0.5%) participants were having sinusitis (Table 1). During the lockdown, 73 (54.5%) male and 159 (53.4%) female students reported having symptoms of anxiety, which reduced to 57 (42.5%) and 142 (47.7%) after the lockdown was lifted. With respect to depression, during the lockdown phase, 75 (56%) males and 199 (66.8%) females had depression, which decreased to 58 (43.3%) and 147 (49.3) after the lockdown (Table 3).

## 3.2 Association of anxiety and depression with participants' demographics

To evaluate the effect of lockdown on the mental health of students, McNemar's test was ran to assess the differences during and after COVID-19 lockdown. As shown in Table 2, both outcome measures showed a significant improvement from during lockdown to after lockdown (anxiety, p<0.001 & depression, p<0.001). Anxiety during COVID-19 lockdown (53.7%) has improved in post-lockdown phase (46.1%). Depression has also decreased from during the COVID-19 lockdown (63.4%) to post-lockdown (47.5%).

## 3.3 Determinants of anxiety and depression

Pearson's Chi-square test (Table 3) aimed to identify the association between participants' characteristics with anxiety and depression. During the lockdown, family income [$\chi^2$ (1, n = 124) = 5.155, p = 0.023] and PA [$\chi^2$ (1, n = 134) = 6.366, p = 0.012] were found to be significantly associated with anxiety. Students with family income in the B40 group (59%) presented with an increased level of anxiety than students in the higher income group (48%), while inactive students (49%) were less likely to be anxious than active students (62%). Depression was significantly associated with gender [$\chi^2$ (1, n = 75) = 4.655, p = 0.031], where male students (56%) were less likely to be depressed when compared to female students (69%).

After the COVID-19 lockdown, family income [$\chi2$ (1, n = 111) = 8.089, p = 0.004] was found to be significantly associated with anxiety, where students with family income in the B40 group (53%) had more odds of being anxious than students with high family income (40%). Depression was significantly associated with family income [$\chi2$ (1, n = 115) = 9.305,

**Table 2. Difference in anxiety, depression, and physical activity level before and after COVID-19 lockdown.**

|  | During Lockdown | Post-Lockdown | Mcnemar's Test |
|---|---|---|---|
|  | n (%) | n (%) | p-value |
| **Anxiety** |  |  | <0.001** |
| Yes | 232 (53.7) | 199 (46.1) |  |
| No | 200 (46.3) | 233 (53.9) |  |
| **Depression** |  |  | <0.001** |
| Yes | 274 (63.4) | 205 (47.5) |  |
| No | 158 (36.6) | 227 (52.5) |  |

McNemar's Test, level of significance <0.05

**Table 3. Association of anxiety and depression with participant characteristics.**

| | During Lockdown | | After Lockdown | |
|---|---|---|---|---|
| | Anxiety | | | |
| | Yes | None | Yes | None |
| | n (%) | n (%) | n (%) | n (%) |
| Age | | | | |
| 18–21 | 181 (52.6) | 163 (47.4) | 156 (45.3) | 188 (54.7) |
| 22–26 | 51 (58.0) | 37 (42.0) | 43 (48.9) | 45 (51.1) |
| $\chi^2$ | 0.803 | | 0.348 | |
| p-value | 0.370 | | 0.555 | |
| Gender | | | | |
| Male | 73 (54.5) | 61 (45.5) | 57 (42.5) | 77 (57.5) |
| Female | 159 (53.4) | 139 (46.6) | 142 (47.7) | 156 (52.3) |
| $\chi^2$ | 0.0472 | | 0.973 | |
| p-value | 0.829 | | 0.324 | |
| Living condition | | | | |
| Living W Others | 21 (61.8) | 13 (38.2) | 22 (46.8) | 25 (53.2) |
| Living W Family | 211 (53.0) | 187 (47.0) | 177 (46.0) | 208 (54.0) |
| $\chi^2$ | 0.965 | | 0.012 | |
| p-value | 0.326 | | 0.914 | |
| Living Area | | | | |
| Rural | 49 (53.8) | 42 (46.2) | 39 (47.6) | 43 (52.4) |
| Urban | 183 (53.7) | 158 (46.3) | 160 (45.7) | 199 (54.3) |
| $\chi^2$ | 0.001 | | 0.091 | |
| p-value | 0.976 | | 0.763 | |
| Restriction level | | | | |
| Complete restriction | 132 (56.9) | 100 (43.1) | 27 (56.2) | 21 (43.8) |
| Other restriction | 100 (50.0) | | 172 (44.8) | 212 (55.2) |
| $\chi^2$ | 2.055 | 100 (50.0) | 2.255 | |
| p-value | 0.153 | | 0.133 | |
| Family income | | | | |
| B40 | 124 (59.3) | 85 (40.7) | 111 (53.1) | 98 (46.9) |
| M40/T20 | 108 (48.4) | 115 (51.6) | 88 (39.5) | 135 (60.5) |
| $\chi^2$ | 5.155 | | 8.089 | |
| p-value | 0.023** | | 0.004** | |
| Pre-lockdown habits | | | | |
| No | 92 (52.9) | 82 (47.1) | 86 (49.4) | 88 (50.6) |
| Yes | 140 (54.3) | 118 (46.3) | 113 (43.8) | 233 (53.9) |
| $\chi^2$ | 0.081 | | 1.324 | |
| p-value | 0.776 | | 0.250 | |
| IPAQ-SF | | | | |
| Inactive | 134 (49.1) | 139 (50.9) | 103 (45.0) | 126 (55.0) |
| Active | 98 (61.6) | | 96 (47.3) | |
| $\chi^2$ | 6.366 | 61 (38.4) | 0.232 | 107 (52.7) |
| p-value | 0.012** | | 0.630 | |
| | Depression | | | |
| | Yes | None | Yes | None |
| | n (%) | n (%) | n (%) | n (%) |
| Age | | | | |
| 18–21 | 216 (62.8) | 128 (37.2) | 159 (46.2) | 185 (53.8) |
| 22–26 | 58 (65.9) | | 46 (52.3) | |

**Table 3.** (Continued)

| | During Lockdown | | After Lockdown | |
|---|---|---|---|---|
| χ2 | 0.294 | 30 (34.1) | 1.029 | 42 (47.7) |
| p-value | 0.588 | | 0.310 | |
| Gender | | | | |
| Male | 75 (56.0) | 59 (44.0) | 58 (43.3) | 76 (56.7) |
| Female | 199 (66.8) | 99 (33.2) | 147 (49.3) | 151 (50.7) |
| χ2 | 4.655 | | 1.355 | |
| p-value | 0.031** | | 0.244 | |
| Living condition | | | | |
| Living W Others | 26 (76.5) | 8 (23.5) | 27 (57.4) | 20 (42.6) |
| Living W Family | 248 (62.3) | 150 (37.7) | 178 (46.2) | 207 (53.8) |
| χ2 | 2.707 | | 2.112 | |
| p-value | 0.100 | | 0.146 | |
| Living Area | | | | |
| Rural | 55 (60.4) | 36 (39.6) | 41 (50.0) | 41 (50.0) |
| Urban | 219 (64.2) | 122 (35.8) | 164 (46.9) | 186 (53.1) |
| χ2 | 0.443 | | 0.263 | |
| p-value | 0.506 | | 0.608 | |
| Restriction level | | | | |
| Complete restriction | 152 (65.5) | 80 (34.5) | 28 (58.3) | 20 (41.7) |
| Other restriction | 122 (61.0) | 78 (39.0) | 177 (46.1) | 207 (53.9) |
| χ2 | 0.945 | | 2.563 | |
| p-value | 0.331 | | 0.109 | |
| Family income | | | | |
| B40 | 135 (64.6) | 74 (35.4) | 115 (55.0) | 94 (45.0) |
| M40/T20 | 139 (62.3) | 84 (37.7) | 90 (40.4) | 133 (59.6) |
| χ2 | 0.238 | | 9.305 | |
| p-value | 0.626 | | 0.002** | |
| Pre-lockdown habits | | | | |
| No | 111 (63.8) | 63 (36.2) | 90 (51.7) | 84 (48.3) |
| Yes | 163 (63.2) | 95 (36.8) | 115 (44.6) | 143 (55.4) |
| χ2 | 0.017 | | 2.131 | |
| p-value | 0.896 | | 0.144 | |
| IPAQ-SF | | | | |
| Inactive | 172 (63.0) | 101 (37.0) | 111 (48.5) | 118 (51.5) |
| Active | 102 (64.2) | 57 (35.8) | 94 (46.3) | 109 (53.7) |
| χ2 | 0.057 | | 0.203 | |
| p-value | 0.811 | | 0.653 | |

Chi-square test, level of significance <0.05

p = 0.002], and students with low family income (55%) were at higher risk of being depressed than students in families with high-income (40%).

## 3.4 Predictors of anxiety and depression during and after COVID-19 Lockdown

To further assess the predictors of anxiety and depression, logistic regression was carried out. During the lockdown, family income and PA showed to be significant predictors of anxiety in

**Table 4. The predictors of anxiety and depression during and after the lockdown.**

| | During Lockdown | | | | After Lockdown | | | |
|---|---|---|---|---|---|---|---|---|
| | OR | p-value | 95% CI for OR | | OR | p-value | 95% CI for OR | |
| | | | Lower | Upper | | | Lower | Upper |
| **Anxiety** | | | | | | | | |
| **Age** | 0.909 | 0.706 | 0.554 | 1.492 | 0.889 | 0.642 | 0.543 | 1.458 |
| **Gender** | 1.080 | 0.722 | 0.706 | 1.652 | 0.831 | 0.392 | 0.543 | 1.270 |
| **Living condition** | 1.438 | 0.349 | 0.672 | 3.077 | 1.068 | 0.841 | 0.559 | 2.040 |
| **Living Area** | 0.943 | 0.813 | 0.582 | 1.528 | 0.965 | 0.890 | 0.584 | 1.595 |
| **Restriction level** | 1.363 | 0.124 | 0.918 | 2.025 | 1.453 | 0.245 | 0.774 | 2.727 |
| **Family income** | 1.611 | 0.018** | 1.084 | 2.393 | 1.671 | 0.011** | 1.126 | 2.480 |
| **Pre-lockdown habits** | 1.018 | 0.930 | 0.682 | 1.520 | 1.264 | 0.254 | 0.847 | 1.886 |
| **Physical Activity** | 0.589 | 0.011** | 1.084 | 2.393 | 0.845 | 0.411 | 0.565 | 1.263 |
| **Depression** | | | | | | | | |
| **Age** | 0.934 | 0.794 | 0.561 | 1.557 | 0.845 | 0.508 | 0.514 | 1.390 |
| **Gender** | 0.648 | 0.046** | 0.422 | 0.993 | 0.780 | 0.256 | 0.509 | 1.197 |
| **Living condition** | 1.935 | 0.125 | 0.832 | 4.501 | 1.797 | 0.080 | 0.933 | 3.462 |
| **Living Area** | 0.875 | 0.592 | 0.536 | 1.427 | 1.063 | 0.814 | 0.641 | 1.761 |
| **Restriction level** | 1.204 | 0.368 | 0.804 | 1.803 | 1.514 | 0.200 | 0.803 | 2.855 |
| **Family income** | 1.100 | 0.645 | 0.734 | 1.647 | 1.702 | 0.009** | 1.145 | 2.529 |
| **Pre-lockdown habits** | 0.990 | 0.961 | 0.657 | 1.492 | 1.299 | 0.203 | 0.868 | 1.942 |
| **Physical Activity** | 0.937 | 0.761 | 0.618 | 1.421 | 1.079 | 0.711 | 0.720 | 1.617 |

Logistic Regression: Reference for Age (18–21 years old), Gender (Male), Living condition (Living with others), Living Area (Rural), Restriction level (Complete restriction), Family income (B40), Pre-lockdown habits (None), Anxiety (Yes), Depression (Yes), Physical Activity (Inactive) are as indicated.

**Statistics are significant at p<0.05

the model $\chi2(9)$ = 18.255, P = 0.032, variance 5.5% (Nagelkerke R Square). Students in the low family income group were 1.61 times more likely to be anxious and inactive students were 0.59 times less likely to become anxious. For depression, the model was found to be statistically insignificant $\chi2(9)$ = 8.987, P = 0.438, variance 2.8% (Nagelkerke R Square), with gender being the only predictor detected, in which male students were 0.65 times less likely to become depressed than females.

After the lockdown, the regression model showed to be statistically insignificant $\chi2(9)$ = 13.223, P = 0.153, variance 4% (Nagelkerke R Square) for anxiety, with family income being the only predictor where students from the low family income group were 1.67 times more likely to be anxious (Table 4). For depression, family income was detected to be the only significant predictor in the model $\chi2(9)$ = 19.808, P = 0.019, variance 6% (Nagelkerke R Square). It was found that students in the low family income group had 1.702 more odds of being depressed.

## Discussion

To the best of our knowledge, this is the first study that has comprehensively explored and compared the determinants and predictors of anxiety and depression negatively impacting the mental health of university students during and after the COVID-19 lockdown. Participants' characteristics that showed significant association with anxiety were family income and PA. In the case of depression, it was gender and family income. During the lockdown, students from low-income group had a higher level of anxiety than the high-income family. Whereas

physically inactive students were less likely to be anxious than active students, and females were more likely to be depressed than males. Similarly, the study conducted among Northern Italian University students also found that females were significantly more susceptible to developing symptoms of depression and anxiety than males, and improved lifestyle habits including physical activity tends to have a protective role against depression [10]. Similar findings were also reported among the university students in Bangladesh, whereof, the prevalence of severe anxiety and depression was higher in women [31]. After lockdown, students from low-income family had more odds of being anxious and depressed than the high-income family. As expected, anxiety and depression improved in the post-lockdown period. We found that the predictors of anxiety were family income and PA, while for depression, the predictors were gender and family income. During lockdown, for anxiety, low family income and being inactive were the predictors, while for depression, female gender was the predictor. After lockdown, low family income was a significant predictor for both anxiety and depression.

Family income can affect the stability of the household and lead to drastic consequences; hence students in the low-income group demonstrated poor mental health (higher anxiety and depression) than those in the high-income group. Among the pandemic stressors, the financial constraint was highlighted as a significant stressor since it kept students worried about the fulfilment of their education due to the loss of family income [32]. Abrupt interruption in studies would have also contributed to the worsening of anxiety and depression since they would be exposed to COVID-related information for most of the time, thus resulting in the accumulation of unsettling thoughts [33]. PA level was associated with and a predictor of anxiety during the lockdown, where inactive students were less likely to be anxious. Generally, exercise has an optimal effect on an individual's mental health; however, the type, duration, and intensity of exercise elicit a different range of responses [34]. It is observed that an exercise regime of 30–60 minutes usually reaps the highest benefits in lowering mental health burdens, and it slowly declines with excessive duration and may negatively impact the mental state when it exceeds 3 hours [34]. This indicates that participants in our study may have been performing exercises for longer durations, which might have elevated their anxiety and depression levels. Our study demonstrated that females were associated with and a predictor of depression during and after the COVID-19 lockdown, and this finding was similar to other studies, where authors found that females face mental health issues during health emergencies and mandatory quarantine [6, 35]. In addition, females capacity for uncertainty is suggested to be lower than males, and this may have led to a significant increase in the mental health scores [32]. We hypothesize this uncertainty among female university students probably comes from being more fearful of COVID-19 than male students, as demonstrated by some other studies [36, 37], and the relatively new area of research would have more insights to unravel with time.

Surprisingly, age, living area, living condition, and restriction level were not associated with either outcome. Age was not a significant determinant of anxiety or depression both during and after the lockdown. This may be due to the fact that students were utilizing electronic gadgets for the online classes, so there were no discrepancies between the younger and older student groups as they were exposed to social media for similar periods of time. For living area i.e. rural or city area and living condition i.e. living with or without their families, were not statistically associated with anxiety or depression either. This could be attributed to the fact that it was not the first lockdown imposed by the Malaysian government. It has been almost two years, ever since the first MCO was implemented, so the students might have possibly adapted to living with MCO. When it comes to the restriction level that students were practicing, there was no association with either anxiety or depression, similar to a study in Germany where higher restriction levels led to increased feelings of loneliness and did not contribute to anxiety or depression symptomatology [38]. This suggests future programs to help mitigate feelings of

loneliness which could assist in combatting anxiety and depression. Pre-lockdown habits were not associated with anxiety and depression of students both during and after the lockdown. In China, no changes were observed in the PA level of 60% Chinese adults and this could explain the insignificant differences in scores for anxiety and depression since their daily routine was probably similar to that before the lockdown [39]. Students who have practiced PA before lockdown would continue to do so entering the lockdown, and their mental health status would remain the same and vice versa.

There were noticeable improvements in the anxiety and depression levels while comparing the during and after COVID-19 lockdown timelines, which signifies that the lockdown has had a significant negative impact on the mental health of university students, similar to a study in Italy where the ill-effects of the lockdown quickly vanished after it was lifted [40]. There has been a significant reduction in anxiety after the lockdown [40], and this should not be a surprise since students are no longer confined to limited spaces, and are readily engaged in activities and entertainment. Social distancing is no longer necessary; gyms, swimming pools, theme parks and local recreations parks are no longer prohibited, and students are now able to develop and maintain interpersonal contact and relationship [33]. Positive effect was seen in people who leave their home 3 or more times per week compared to those leaving less often and this was attributed to the increased variation in the routine of the people [41]. Being able to be free, meet people, and experience the novel certainly brings positivity and excitement and improves the mental health among students. Our study demonstrated that compared to the lockdown period, the mental health state of students improved after the lockdown was lifted, which was promising.

Theoretically, the study contributed to ample scientific knowledge in the growing field of COVID-19-related research by identifying the determinants of poor mental health among young adults. Thus it will lead to effective mental health education, promotion, and supportive measures, especially for the at-risk population. Practical implications would focus on initiating feasible problem-solving methods to develop a resilient post-pandemic recovery program. Since the study findings indicated that family income is a significant predictor for anxiety and depression; hence university management, relevant governing authorities, and policymakers should target underprivileged students from lower socioeconomic backgrounds and provide them with minimum financial incentives. In addition, to improve the overall psychological well-being of vulnerable students, intervention strategies such as resilience programs can be incorporated in the recovery plan to strategize effective coping skills among students [42]. In summary, the study findings can be applied extensively in tertiary educational settings by implementing awareness campaigns, extracurricular activities, and female-centric social support groups for promoting sound mental health in young adults.

## 4.2 Limitations and future recommendations

Data was collected online through Google forms using self-reporting questionnaire, so there might be possibility of biasness arising from social desirability. We would recommend conducting future studies in face-to-face fashion which could increase the credibility of the response. In addition, the retrospective nature of the study contributed to some notable shortcomings and led to recall bias. While obtaining the data for "during COVID-19 lockdown", it was done in a recalling fashion since the data collection process started when the lockdown has already been lifted, hence, the recall bias was inevitable. This may be subjected to some confounding effect and may influence the temporal relationship between students' mental health status and the COVID-19 lockdown. Therefore, extensive research is required prospectively or using a longitudinal method to account for the recall bias in future studies on global emergency.

## Conclusion

During the lockdown, students from low-income families experienced higher anxiety than the high-income groups, inactive students were less likely to be anxious than active students, and female students were more likely to be depressed compared to male students. After the lockdown, students from low-income families had more odds of being anxious and depressed than the high-income families. As expected, anxiety and depression improved post-lockdown, highlighting that the lockdown negatively affected the mental health of university students. During the lockdown, low family income and physically inactive status were the predictors for anxiety; while for depression, gender (female) was the predictor. In short, certain groups of students were highly vulnerable than others during and after the lockdown. Thus, there is a strong need for implementing public health measures based on the significant findings of this study and facilitating vulnerable students with better coping mechanisms. Monetary assistance, resilience interventions, and active lifestyle programs might serve as effective, supportive strategies for promoting sound mental health in young individuals.

## Supporting information

**S1 Checklist. STROBE statement—Checklist of items that are included in this observational study.**
(DOCX)

## Author Contributions

**Conceptualization:** Imtiyaz Ali Mir.

**Data curation:** Imtiyaz Ali Mir.

**Formal analysis:** Imtiyaz Ali Mir, Muhammad Noh Zulfikri Mohd Jamali, Mohammed AbdulRazzaq Jabbar.

**Investigation:** Imtiyaz Ali Mir, Shang Kuan Ng.

**Methodology:** Imtiyaz Ali Mir, Shang Kuan Ng, Muhammad Noh Zulfikri Mohd Jamali.

**Supervision:** Imtiyaz Ali Mir.

**Writing – original draft:** Shang Kuan Ng, Syeda Humayra.

**Writing – review & editing:** Imtiyaz Ali Mir, Muhammad Noh Zulfikri Mohd Jamali, Syeda Humayra.

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
