## [Decision Letter · Decision Letter 0]

11 Nov 2022

PONE-D-22-22931Determinants of Mental Health during and after COVID-19 Lockdown among University Students in MalaysiaPLOS ONE

Dear Dr. Mir,

Thank you for submitting your manuscript to PLOS ONE. After careful consideration, we feel that it has merit but does not fully meet PLOS ONE’s publication criteria as it currently stands. Therefore, we invite you to submit a revised version of the manuscript that addresses the points raised during the review process.

Two expert reviewers have provided their comments that can be used to improve the work. I also agree with them that the present contribution has merits; however, revisions are needed. Reviewer #2 has provided some references that are relevant to the present contribution, and the authors can consider using them. I would also encourage the authors consider some other papers on university students' mental health:Sharma, R., Bansal, P., Chhabra, M., Bansal, C., & Arora, M. (2021). Severe acute respiratory syndrome coronavirus-2-associated perceived stress and anxiety among Indian medical students: A cross-sectional study. Asian Journal of Social Health and Behavior, 4, 98-104.Ahorsu, D. K., Pramukti, I., Strong C., Wnag, H.-W., Griffiths, M. D., Lin, C.-Y., Ko, N.-Y. (2021). COVID-19-related variables and its association with anxiety and suicidal ideation: Differences between international and local university students in Taiwan. Psychology Research and Behavior Management, 14, 1857-1866. Nayan, M. I., Uddin, M. S., Hossain, M. I., Alam, M. M., Zinnia, M. A., Haq, I., Rahman, M. M., Ria, R., Haq Methun, M. I. (2022). Comparison of the performance of machine learning-based algorithms for predicting depression and anxiety among University Students in Bangladesh: A result of the first wave of the COVID-19 pandemic. Asian Journal of Social Health and Behavior, 5, 75-84.Pramukti, I., Strong, C., Sitthimongkol, Y., Setiawan, A., Pandin M. G. R., Yen, C.-F., Lin, C.-Y., Griffiths, M. D., Ko, N.-Y. (2020). Anxiety and suicidal thoughts during the COVID-19 pandemic: A cross-country comparison among Indonesian, Taiwanese, and Thai university students. Journal of Medical Internet Research, 22(12), e24487.Also, it would be good if the authors can have some references to indicate effective programs in improving university students' mental health.Kadian, S., Joseph, J., Pal, S., & Devi, R. (2022). Brief resilience interventions for mental health among college students: Randomized controlled trial. Asian Journal of Social Health and Behavior, 5, 131-137.

We look forward to receiving your revised manuscript.

Kind regards,

Chung-Ying Lin

Academic Editor

PLOS ONE

Journal Requirements:

Reviewers' comments:

Reviewer's Responses to Questions

**Comments to the Author**

1. Is the manuscript technically sound, and do the data support the conclusions?

Reviewer #1: Yes

Reviewer #2: Yes

2. Has the statistical analysis been performed appropriately and rigorously? 

Reviewer #1: Yes

Reviewer #2: Yes

3. Have the authors made all data underlying the findings in their manuscript fully available?

Reviewer #1: Yes

Reviewer #2: Yes

4. Is the manuscript presented in an intelligible fashion and written in standard English?

Reviewer #1: Yes

Reviewer #2: Yes

5. Review Comments to the Author

Reviewer #1: This is, in summary, an interesting study aimed to explore and compare the determinants and predictors of mental health (anxiety and depression) during and after COVID-19 lockdown among university students. The authors reported that during lockdown, family income, and physical activity were associated with anxiety, while depression was associated with gender. They also added that after lockdown, family income was found to be associated with both anxiety, and depression. Moreover, during lockdown, family income, and physical activity were predictors for anxiety, and gender being the only predictor for depression. Finally, after lockdown, family income was a predictor for both anxiety, and depression.

The authors may find my minor comments below.

First, when throughout the Introduction section, the authors correctly stressed the importance and complexity of transition into adulthood of university students, the impact of brain changes in early-onset individuals with mood disorders, particularly among young adolescents for their possible effects on childhood negative experiences, might be briefly mentioned. Importantly, more abnormalities have been documented in the brains of adolescents with bipolar depression than unipolar depression. Therefore, given the above information, my suggestion is to include the following manuscripts (PMID: 25212880; 35645550; 33417221).

In addition, the authors might further mention the consequences of Covid-19 related lockdown on lifestyle habits and behavioral risk factors. Importantly, the impact of COVID-19 lockdown on physical, mental, and social wellbeing of elderly and fragile populations in specific countries such as Italy cannot be ignored. The multi-disciplinary competencies together with appropriate funding and access to rich data sources may allow to fulfill interesting research objectives. Thus, according to this background, the following articles (PMID: 32701921; 35886439) may be cited within the main text.

Moreover, as the most relevant aims/objectives of the present review paper have been reported extensively, the main hypotheses underlying this study should be reported in a similarly detailed manner.

In addition, the main rationale according to which the study has been conducted and criteria based on which the online survey has been proposed and participants have been recruited need to be comprehensively described for the general readership.

Moreover, the most relevant outcome measures need to be described more succinctly for the general readership.

The authors could immediately present and discuss, in the first lines of the Discussion section, the most relevant study conclusive remarks of this paper instead of focusing on the most relevant aims/objectives of the present study that should have been adequately stressed elsewhere within the main text.

In addition, the most relevant shortcomings of this paper need to be extensively reported as the main caveats (e.g., the retrospective nature of the study) have been only partially described.

Finally, what is the take-home message of this manuscript? While the authors reported that during lockdown, students from low-income family have higher anxiety than the higher income, and inactive students were less likely to be anxious than active students, whilst female students were more likely to be depressed when compared to male students, they could provide, in my opinion, more extensively conclusive remarks upon the main topic for the readers. What are, specifically, the main implications of these findings? How the present results may be generalized? Here, some additional information are required and might be useful for the general readership.

Reviewer #2: It's a very important topic focused on the effects of the COVID-19 Lockdown on University Students' Mental Health in Malaysia. However, I have minor comments on the manuscript to be accepted.

- In general, the manuscript needs Proofreading and exceptional attention to the consistencies from the introduction to the conclusion.

- Please Provide a STROBE Statement per the cross-sectional study, prepare your manuscript according to STROBE guidelines, submit the checklist as per STROBE criteria, and attach it as a supplementary file.

Abstract;

Please write the abstract as one paragraph, including all the main points for the structured abstract.

Introduction:

- The paper focused on the "Determinants of Mental Health during and after COVID-19 Lockdown among University Students in Malaysia" It is hard to follow the study's main aim because there are inconsistencies between the headline, intro, findings, and discussion.

- Your study focused on the effect of the COVID-19 Lockdown on University Students' Mental Health in Malaysia. However, you didn't concentrate on or highlight this main issue and what previous studies found on these issues in the introduction. Therefore, It is recommended to extend the introduction part by including how the COVID-19 Lockdown affects University Students' Mental Health and education as well globally and in Malaysia specifically; here are some studies focused on these issues, and you may find more, and later on you may use in addition to the introduction in your discussion as well to compare with your results.

- https://doi.org/10.1371/journal.pone.0277368

- https://doi.org/10.3390/healthcare10101858

- https://doi.org/10.1016/j.sleepe.2022.100030

- https://doi.org/10.3389/feduc.2022.960660

- https://doi.org/10.3390/ejihpe12080079

Methods:

- The subsection "Role of funders" is not under the method; please move it and follow the PLOS ONE GUIDELINE.

- Under the methods, please include a subsection about"Ethics and consent statement"

Results:

Please report the sociodemographic characteristics of the sample clearly, and please take note "97% were Chinese" they are Malaysian but their ethnicity is Chinese, and the study was conducted in Malaysia, and you cant start the sentence with a number; please rewrite.

- Please report the level of anxiety and depression among gender and other sociodemographic factors in the table and the text.

Discussion:

In the discussion, please highlight the theoretical and practical implications.

6. PLOS authors have the option to publish the peer review history of their article (what does this mean?). If published, this will include your full peer review and any attached files.

Reviewer #1: No

Reviewer #2: **Yes: **Musheer A. Aljaberi

---

## [Author Response · Author response to Decision Letter 0]

28 Dec 2022

Dear Academic Editor & Reviewers,

We appreciate the time and effort you have dedicated to provide the feedback on our manuscript and are grateful for the valuable and insightful comments to help us improve the quality of our paper. We have incorporated most of the suggestions made by both academic editor and reviewers, those changes are highlighted within the manuscript. Our response to your comments is provided in the rebuttal letter which is attached as a supplementary file. We do hope that the academic editor and reviewers’ find this manuscript acceptable for publication.

---

## [Editor Report · Decision Letter 1]

2 Jan 2023

Determinants and Predictors of Mental Health during and after COVID-19 Lockdown among University Students in Malaysia

PONE-D-22-22931R1

Dear Dr. Mir,

We’re pleased to inform you that your manuscript has been judged scientifically suitable for publication and will be formally accepted for publication once it meets all outstanding technical requirements.

Kind regards,

Chung-Ying Lin

Academic Editor

PLOS ONE

Additional Editor Comments (optional):

The authors have seriously taken all the comments from the expert reviewer and I into consideration. The  authors have carefully reviewed the suggested references for appropriateness and added those relevant references in the revised manuscript, this is satisfactory to a revision. Also, the authors have improved their manuscript quality with other comments provided by the expert reviewer. Therefore, I am happy to accept the manuscript in the present form. I sincerely thank the authors taking such hard efforts in conducting the study and revising the manuscript.  
---

## [Editor Report · Acceptance letter]

6 Jan 2023

PONE-D-22-22931R1 

Determinants and Predictors of Mental Health during and after COVID-19 Lockdown among University Students in Malaysia 

Dear Dr. Mir:

I'm pleased to inform you that your manuscript has been deemed suitable for publication in PLOS ONE. Congratulations! Your manuscript is now with our production department. 

Kind regards, 

on behalf of

Dr. Chung-Ying Lin 

Academic Editor

PLOS ONE